# Marginal Micro-Seal and Tensile Bond Strength of a Biopolymer Hybrid Layer Coupled with Dental Prosthesis Using a Primerless-Wet System

**DOI:** 10.3390/polym15020283

**Published:** 2023-01-05

**Authors:** Morakot Piemjai, Onusa Waleepitackdej, Franklin Garcia-Godoy

**Affiliations:** 1Department of Prosthodontics, Faculty of Dentistry, Chulalongkorn University, Bangkok 10330, Thailand; 2Department of Bioscience Research, College of Dentistry, University of Tennessee Health Science Center, Memphis, TN 38163, USA; 3Adjunct Faculty, The Forsyth Institute, Cambridge, MA 02142, USA

**Keywords:** primerless-wet bonding, resin adhesive system, hybrid layer, tensile bond strength, micro-seal, luting resin, dental prosthesis, fixed prosthodontics

## Abstract

The aim of this study is to compare the marginal seal and tensile bond strength (TBS) of prostheses fixed to enamel-dentin using different adhesive systems. Resin-composite inlays directly fabricated from Class V cavities of extracted human molars/premolars and mini-dumbbell-shaped specimens of bonded enamel-dentin were prepared for microleakage and tensile tests, respectively. Four adhesive systems were used: primerless-wet (1-1 etching for 10-, 30-, or 60-s, and 4-META/MMA-TBB), primer-moist (All-Bond2 + Duolink or Single-Bond2 + RelyX ARC), self-etch (AQ-Bond + Metafil FLO), and dry (Super-Bond C&B) bonding. Dye penetration distance and TBS data were recorded. Failure modes and characteristics of the tooth-resin interface were examined on the fractured specimens. All specimens in 10-, 30-, and 60-s etching primerless-wet, Super-Bond, and AQ-Bond had a microleakage-free tooth-resin interface. Primer-moist groups showed microleakage at the cementum/dentin-resin margin/interface. Significantly higher TBSs (*p* < 0.05) were recorded in primer-less-wet and Super-Bond groups with the consistent hybridized biopolymer layer after the chemical challenge and mixed failure in tooth structure, luting-resin, and at the PMMA-rod interface. There was no correlation between microleakage and TBS data (*p* = −0.148). A 1–3 µm hybrid layer created in the 10–60 s primerless-wet technique, producing complete micro-seal and higher tensile strength than enamel and cured 4-META/MMA-TBB, may enhance clinical performances like Super-Bond C&B, the sustainable luting resin.

## 1. Introduction

Dental enamel naturally protects the dentin and pulp from invasion by external stimuli. Therefore, non- or minimally invasive restorations or prostheses that protect the enamel from tooth reduction, recurrent caries, or tooth fracture are crucial in maintaining healthy dentin and pulp. High tensile bond strength adhesives are required when restorations or prostheses are not sufficiently resistant to displacement under functional loading [1]. Severe tooth reduction to gain more retention, resistance form, or strength for restorations/prostheses removes dentin, especially when restoring with non-hybrid layer formation materials, such as amalgam restorations and dental prostheses fixed with acid-base cement.

The total-etch concept was developed to simplify bonding to both enamel and dentin by etching the entire cavity with 40% phosphoric acid gel [2]. Strong phosphoric acid demineralizes enamel deeper than mild acid [3]. Thus, demineralized enamel might remain after resin polymerization allowing oral acid penetration. However, monomer diffusion into etched enamel is more accessible than demineralized dentin, as phosphoric acid-conditioned dentin collapses when air-dried [4]. Therefore, phosphoric acid demineralized dentin cannot provide adequate permeability for complete monomer impregnation in either dry or moist systems [4,5]. In addition, it leads to a leakage pathway [6,7], post-operative hypersensitivity, and secondary caries [8,9].

Ferric ions in an acid conditioner can aggregate glycosaminoglycan (GAG) in de-mineralized dentin and provide permeability for potential monomers to diffuse through completely in dry or wet conditions [6,7,8,9,10,11]. Therefore, a hybridized dentin with a leakage-free interface was formed [6,7,8,9]. Self-etch bonding systems and self-adhesive cement were introduced to simplify the bonding steps and minimize aggressive phosphoric acid etching on dentin. However, bonded restorations using these self-etching or self-adhesive systems could not reliably provide a leakage-free dentin-resin interface [12,13,14,15] because of the limitation of monomer diffusion through any smear layer into the intact dentin [14,16]. Ferric chloride (1%) in 1% citric acid aqueous conditioner (1-1), a mild acid for smear layer removal, and 4-methacryloyloxyethyl trimellitate anhydride in methyl methacrylate initiated by tri-n-butyl borane resin (4-META/MMA-TBB) can provide reliable hybridized dentin when wet bonding with primer in the long-range periods (10–60 s) of conditioning [10]. A 1–3 µm hybridized dentin layer suggested that 1-1 conditioned dentin was sufficiently permeable for water to evaporate and for monomers to impregnate. Thus, only blot-drying with or without primer (primerless-wet bonding) can produce a complete hybrid layer that reinforces the dentin [10,17] and prevents dye penetration of direct restorations [7,17].

Dental clinical failures are often found in direct or indirect restorations and fixed partial prostheses due to secondary caries, especially at the cementum/dentin margin [18,19,20,21]. Detachment of restorations or prostheses is a minor complication that leads to failure [21,22,23,24]. The demineralized dentin, the defect, remains in restored-dentin, which may lead to a leakage pathway and recurrent caries [6,8,9], strongly influencing the failure of restorations or prostheses [18,19,20,21]. The hybrid layer formed by dry bonding using 10% citric acid and 3% ferric chloride aqueous solution (10-3) conditioned for 10 s and 4-META/MMA-TBB resin (Super-Bond C&B, Sun Medical, Shiga, Japan; a sustainable luting resin since 1983), provides a significantly higher 15-year survival with less secondary caries and prosthesis detachment complication rates of full coverage retainers than those of acid-base cement [21]. However more extended 10-3 etching period of 30–60 s creates demineralized dentin too deep (>4 µm) to be fully impregnated by the monomers before starting polymerization. Thus, exposed demineralized dentin remains to allow leakage, caries, or pulp infection [8,21]. A tensile test using a mini-dumbbell-shaped bonded specimen [2,25] and a microleakage test [6,8] can detect this remaining demineralized dentin, the weakening part of the restored dentin.

We hypothesized that primerless-wet bonding could create a reliable hybrid layer on enamel-dentin and provide a complete micro-seal and tensile bond strength comparable with Super-Bond C&B. Moreover, a complete seal might not relate to the tensile bond strength of tooth-resin interface luting with various resin adhesives.

The objective of this study was to compare the dye penetration distance and the tensile bond strength at the tooth-resin interface of a prosthesis fixed to enamel-dentin using different adhesive systems: dry (Super-Bond C&B), moist with primer (All-Bond2 + Duoink or Single-Bond2 + RelyX ARC), self-etch (AQ-Bond + Metafil FLO), and primerless-wet (1-1 conditioner and 4-META/MMA-TBB resin) bonding.

## 2. Materials and Methods

Previously frozen extracted human molars and premolars without caries, restorations, or cracked lines were collected and stored in water at −20 °C for 2–3 months. Then, all teeth were randomly divided into two experimental groups of 7 premolars and 14 molars for micro-seal evaluation and 42 molars to prepare the mini-dumbbell-shaped specimens for tensile testing. The primary experimental steps are illustrated in Figure 1.

### 2.1. Micro-Seal Evaluation Using Dye Penetration

Class V cavities at the cementoenamel junction (CEJ) on the buccal and lingual surfaces of seven premolars and all axial surfaces for fourteen molars were outlined. A box cavity of 2 × 3 mm and 1.5 mm depth with approximately 5° divergent axial walls was prepared with occlusal and gingival margins on enamel and cementum, respectively, using a diamond bur with an air-water sprayed high-speed handpiece. Resin composite inlays of 2 × 3 × 1.5 mm were directly fabricated from the cavities with light-cured resin composite (Metafil CX, Sun Medical, Shiga, Japan). Each inlay was light-cured for 60 s on both outer and inner surfaces. All cavities were randomly divided into 7 groups of 10 specimens (1 premolar and 2 molars) for different tooth conditionings and/or resin cement. Primerless-wet bonding using 1-1 conditioning for 10 s, 30 s, 60 s (Groups 1-1-10s, 1-1-30s, 1-1-60s respectively) and 4-META/MMA-TBB resin; and commercially available adhesive resin cement: Super-Bond C&B (Sun Medical, Shiga, Japan), All-Bond2 + Duolink (Bisco, Schaumburg, IL, USA), Single-Bond2 + RelyX ARC (3M ESPE, Saint Paul, MN, USA), or AQ-Bond Plus + Metafil FLO (Sun Medical, Shiga, Japan) was used to fix an inlay prosthesis into the cavity. The manipulation of commercial systems followed manufacturers’ recommendations, as shown in Table 1, and the main chemical composition of luting adhesives and resin composite inlay, as shown in Table 2. Fine diamond burs in a high-speed handpiece were used to finish the restored margins after the polymerization of adhesives. After storing in water at 37 °C for 24 h, all tooth surfaces except an area of the inlay and 1 mm away from the occlusal (enamel) and gingival (cementum) margins were coated with two layers of nail varnish (Pias, Bangkok, Thailand). Specimens were then immersed in 0.5% basic fuchsin dye for 24 h. After soaking, all specimens were cleaned with tap water before being vertically sectioned at the center of each restoration using a diamond disc with a slow-speed handpiece. The distance of dye penetration was measured under a stereomicroscope (ECLIPSE E400 POL, Nikon, Japan) at ×50–×200 magnifications.

### 2.2. Tensile Bond Strength Test

Forty-two extracted sound human molars without cracks were root-embedded in acrylic blocks (Formatray, Kerr, Orange, CA, USA). A 2 mm occlusal portion was horizontally removed using a sectioning machine (Isomet 1000 series 15, Buehler, Lake Bluff, IL, USA) to expose a surface which was then ground with a wheel diamond bur (111 Intensiv, Grancia, Switzerland). A prepared surface of 2 mm in width (0.5 mm of enamel/DEJ and 1.5 mm of dentin) and 4 mm in length was outlined with double-sided tape. One of the tooth conditionings and adhesive systems, as previously mentioned in the micro-leakage test (Table 1), was randomly selected to bond that area with a square PMMA rod (7 × 7 × 4 mm) to form a handle for tensile testing. A 2.0-mm thick vertical section was prepared using the sectioning machine. A mini-dumbbell bonded specimen with a cross-section of 3.0 × 2.0 mm was shaped using a diamond fissure bur (B11, GC Dental Industrial Co., Tokyo, Japan) operated in a high-speed handpiece under the air-water spray. All specimens were stored in 37 °C water for 24 h prior to tensile testing (n = 6) [2,10,16]. Each mini-dumbbell specimen was securely bonded to disposable PMMA jigs using 1-1-10s bonding on the tooth surface and self-cured acrylic (Unifast, Trad, GC Int. Co., Tokyo, Japan) on the PMMA surface to facilitate tensile testing [16]. An assembled specimen was aligned in a universal testing machine (Instron 8872, Norwood, MA, USA) and vertically loaded in tension at a crosshead speed of 1.0 mm/min. The force at failure was recorded in Newtons. The mode of failure, the cross-sectional area of the fractured surface, and the enamel and dentin area were examined under a stereomicroscope and SEM. Tensile bond strengths were calculated in MPa.

### 2.3. Characteristics Evaluation of Tooth-Resin Interfacial Biopolymer Layer

Fracture specimens from each bonding system were randomly selected and vertically sectioned (without epoxy embedding) into 1 mm thick pieces. The tooth-resin interface surface to be examined was finished with #600 and #1200 grit abrasive papers and finally polished with 0.05 µm alumina paste and then ultrasonically cleaned for 15 min. The chemical challenge, either soaking in 6 mol/L HCl for 30 s or soaking in 6 mol/L HCl for 30 s followed by 1% NaOCl for 60 min, was carried out to test the resistance of acidic and proteolytic degradation, akin to caries formation. For SEM examination, all polished and chemically soaked specimens were desiccated and gold-sputtered. The characteristics of the newly formed interfacial biopolymer layer between the tooth and cured resin were examined at ×35 to ×5000 magnifications.

### 2.4. Statistical Analysis

Normal distribution and homoscedasticity of dye penetration distance and tensile bond strength data were analyzed using one-sample Kolmogorov-Smirnov and Levene tests, respectively. In addition, Pearson correlation between leakage distance and tensile bond strength data was performed using SPSS for Windows version 22 (IBM Corporation, Somers, NY, USA). The significant difference was set at α = 0.05.

## 3. Results

Means and standard deviations (SD) of dye penetration distance, tensile bond strength, and mode of failure for all groups are summarized in Table 3. No dye penetration at the cementum/dentin-resin interface was found in the primerless-wet groups (1-1-10s, 1-1-30s, 1-1-60 s) (Figure 2), Super-Bond C&B (Figure 3a), and AQ-Bond (Figure 3b) specimens and at the enamel-resin interface in all groups. No statistically significantly different dye penetration distance at the dentin-resin interfaces was found between All-Bond2 and Single-Bond2 when analyzed using a t-test. All specimens in these moist bonding with primer groups leaked at the dentin-resin interface (Figure 4).

As a significant difference was found in the test of homogeneity of variances, Brown-Forsythe and Tamhane multiple comparisons were used to reveal a significant difference in tensile bond strength between groups. No significant difference in tensile bond strength was found among 1-1-10s, 1-1-30s, 1-1-60s, Super-Bond, and Single-Bond2; Single-Bond2, All-Bond2, and AQ-Bond groups. Cohesive failure originated in enamel followed by either dentino-enamel junction (DEJ), dentin, cured luting resin or adhesive failure at resin-PMMA rod interfaces mainly occurred in fractured specimens of primerless-wet and Super-Bond groups (Figure 5). In contrast, failure occurring in demineralized dentin or at the resin-demineralized dentin interface was found in Single-Bond2 (Figure 6a), and All-Bond2 fractured specimens (Figure 6b). The lowest tensile bond strength was measured in AQ-Bond specimens, where the original failure was found at the suspended resin-smear layer of the enamel-resin interface (Figure 6c).

A consistent thickness of hybridized enamel or hybridized dentin after loading and the chemical challenge was found in primerless-wet (Figure 7) and dry bonding (Super-Bond C&B) (Figure 8) systems. A detached or degraded enamel- or dentin-resin interfacial layer was found in moist with primer (All-Bond2 and Single-Bond2) (Figure 9) and self-etch (AQ-Bond) (Figure 10) systems. The correlation between the dye penetration distance and the tensile bond strength data for the enamel and dentin-bonded interface was very weak (Pearson correlation = −0.148)

## 4. Discussion

The complete marginal seal, no significant differences in TBSs, and the same failure mode among primerless-wet and Super-Bond groups suggest that the milder acid of 1-1 conditioner using primerless-wet bonding could adequately prepare the etched enamel-dentin for 4-META/MMA-TBB resin to entirely impregnate as well as that of the 10-3 conditioner in dry bonding (Figure 2, Figure 3a and Figure 5, Table 3). Furthermore, long etching periods of 10–60 s of 1-1 dissolved less content of calcium ions, therefore even blot-drying without primer could provide the permeability of acid-etched enamel-DEJ-dentin for 4-META/MMA-TBB to penetrate completely before being polymerized to form a 1–3 µm hybrid layer. Therefore, no adhesive failure at the tooth-resin interface was noticed with the average strength like dry bonding using 10-3 solution for 10 s etching of Super-Bond C&B.

The mode of failure originating on the enamel surface suggests that resin infiltration into acid-etched enamel-DEJ-dentin using primerless-wet bonding and dry bonding using Super-Bond C&B could provide a tensile bond strength higher than that of the tensile strength of enamel itself (Figure 5). The complete hybridization of resin into the total etched enamel-DEJ-dentin depends on the demineralized tooth substrate’s permeability and the monomers’ diffusion potential. Non-detachment with consistent thickness hybridized layers against loading force for failure and chemical challenge found in the primeless-wet and Super-Bond groups (Figure 7 and Figure 8) suggest the high resin content encapsulates the tooth component in the hybrid layer. Therefore, the enamel- and dentin-resin hybrid layer, created using a primerless-wet bonding with 10 s to 60 s 1-1 conditioning, 4-META/MMA-TBB, and PMMA powder could be a sustainable biopolymer to provide a complete micro-seal and high tensile bond strength comparable with that of Super-Bond C&B. The long-range conditioning period of 1-1 for 10 s to 60 s ensures more safety manipulation in the clinical situation.

The adhesive failure at the demineralized dentin-resin interface or cohesive failure in the remaining demineralized dentin found in Single-Bond2 and All-Bond2 fractured specimens minimized the tensile bond strength and was probably the cause of the leakage (Figure 4 and Figure 6a,b). This demineralized dentin is the leakage pathway for dye or lactic acid to penetrate [6,8,9]. After tensile loading and chemical challenge, the in-consistent enamel-resin interface and the detached and degraded dentin-resin interface confirmed monomers’ incomplete impregnation into the demineralized tooth substrate (Figure 8). These results suggest that moist bonding using 32% or 35% phosphoric acid for a 15 s etching period, kept moist and either primed and bonded using one or separate steps cannot provide a complete marginal seal of cementum/dentin and a stable hybrid layer.

Although achieving a complete seal for the enamel and cementum/dentin margin/interface (Figure 3b), AQ-Bond specimens had a significantly lower tensile strength than the primerless-wet and Super-Bond groups. The fracture mode originated in the hybridized suspended smear layer of the enamel-resin interface (Figure 6c); the thin hybridized enamel with degradation and the detached hybridized dentin after chemical immersion (Figure 10) suggest the remaining smear and the low resin content of the hybrid layer. These results imply that scrubbing this self-etch monomer for 20 s could not sufficiently remove all the smear layer to provide high adhesion to enamel and dentin. Therefore, careful removal of more smear layers by aggressively air-blowing off or an additional scrubbing application [9] is recommended for cavities with no retentive form and require higher retention, such as a large wedge shape abrasion lesion.

As the primer and bonding agents of all groups contain the methacrylate monomers with hydrophobic and hydrophilic groups, the significantly different factor is the conditioned tooth surface of each system. This study’s results suggest that the permeability of conditioned tissue of the adhesive system that provides the durable biopolymer hybridized dentin influences the complete micro-seal and higher tensile bond strength. Moreover, the complete micro-seal or dye penetration distance was unrelated to the TBS data. Therefore, luting resin or resin adhesives that provide a complete marginal seal should be primarily considered to protect dentin and pulp for long-term function. In other words, a complete seal margin with an impermeable hybrid layer is more reliable than a high tensile bond strength adhesive with the leaked margin in preventing recurrent or secondary caries [8,9,21], the most common dental restoration failure, ensuring the lifelong survival of restored vital teeth. In clinical cases where high retention and completely sealed dentin is required, i.e., a short clinical crown or severe tooth wear and partial coverage retainers, a complete hybrid layer with high tensile strength and micro-seal margins can extend the long-term survival of vital teeth with less invasive treatment or without intentional pulp removal [21,26,27]. The results of this study support the hypothesis.

The novelty of this study is that a primerless-wet system using mild acid (1-1) conditioning for 10–60 s and blot-drying to remove all smears and water is less aggressive and safer than a dry bonding system using a 10-3 conditioner. Furthermore, its total etching creates durable hybridized enamel and dentin, providing the micro-seal and tensile bond strength (TBS) better than a primer-moist system. In addition, its TBS is higher than the self-etch system. However, an in-vivo study should be carried out to evaluate the effect of dentinal fluid in a vital tooth before introducing this system into the market. In the future, dentists can use this adhesive system as long-term dentin protection to treat patients at home.

## 5. Conclusions

Primerless-wet bonding using 1-1 conditioning for 10 s to 60 s and 4-META/MMA-TBB luting resin provided a reliable hybrid layer, a biopolymer, with a marginal micro-seal and tensile strength of the bonded enamel/DEJ/dentin similar to that of a dry system using Super-Bond C&B and higher than that of enamel itself. It can be a sustainable luting resin or adhesive agent with a sustainable hybrid layer. A basic fuchsin dye penetration was found when demineralized cementum/dentin was left underneath to provide a leakage pathway. To successfully prevent biological failure, a luting resin providing a complete marginal seal is preferable to the one with a leaked margin, even with high bond strength, as there is no correlation between complete marginal micro-seal and TBS data.

## Figures and Tables

**Figure 1 polymers-15-00283-f001:**
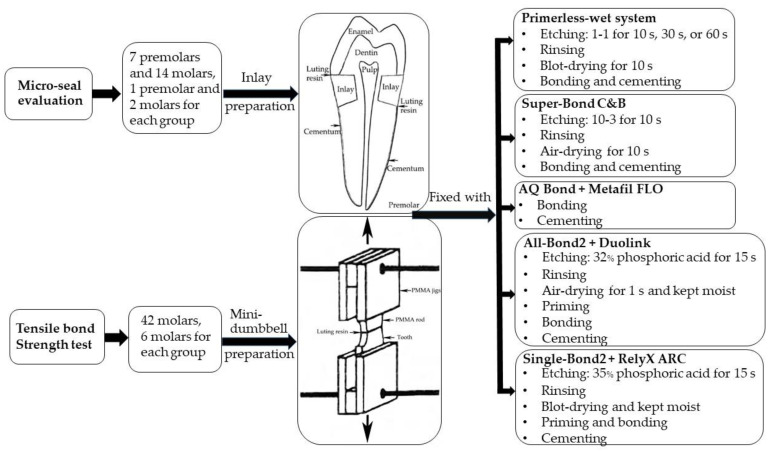
An illustrated diagram for the steps carried out in this experiment.

**Figure 2 polymers-15-00283-f002:**
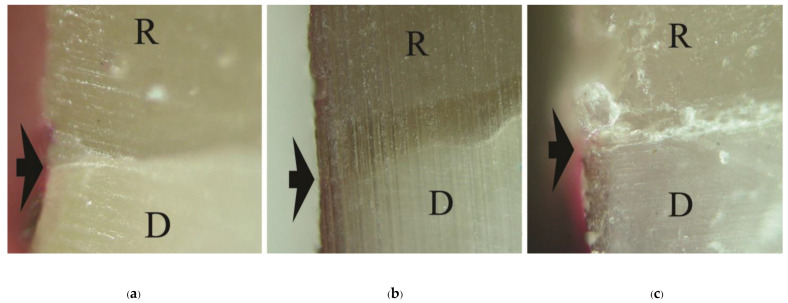
No dye penetration at the cementum/dentin-resin interface (arrowed) of primerless-wet bonding groups: (**a**) 1-1-10s, (**b**) 1-1-30s, (**c**) 1-1-60s (original ×200, D = dentin, R = resin-composite in-lay).

**Figure 3 polymers-15-00283-f003:**
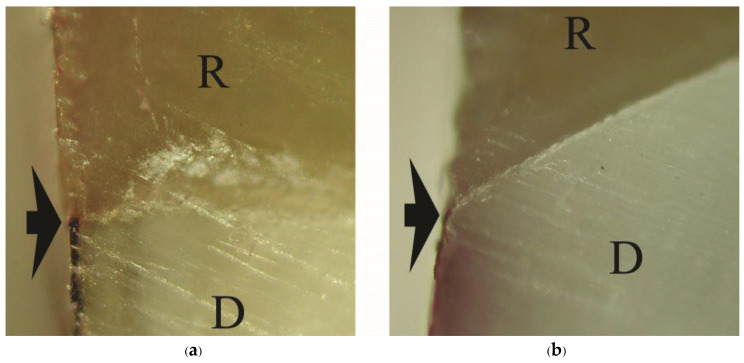
No dye penetration at the cementum/dentin-resin interface (arrowed) of Super-Bond (**a**) and AQ-Bond (**b**) specimens (original ×200, D = dentin, R = resin-composite inlay).

**Figure 4 polymers-15-00283-f004:**
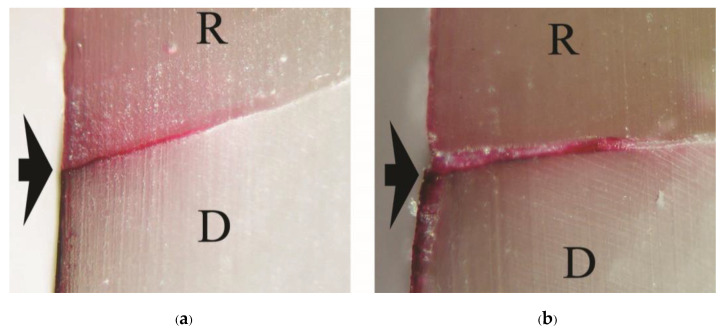
Dye penetration at the cementum/dentin-resin interface (arrowed) of moist bonding with primer groups: (**a**) All-Bond2, (**b**) Single-Bond2 (original ×200, D = dentin, R = resin-composite inlay).

**Figure 5 polymers-15-00283-f005:**
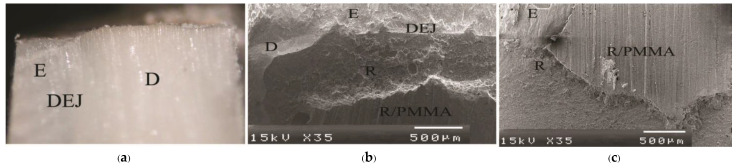
Stereo and SEM micrograph of the fractured surface showing cohesive failure originating in enamel followed by either DEJ, dentin, cured resin, or adhesive failure at resin-PMMA rod interfaces (R/PMMA) primarily found in primerless-wet and Super-Bond groups: sagittal view at ×50 magnification (**a**) and cross-sectional view of 1-1-60s (**b**) and Super-Bond (**c**) specimens (D = dentin, E = enamel, R = luting resin).

**Figure 6 polymers-15-00283-f006:**
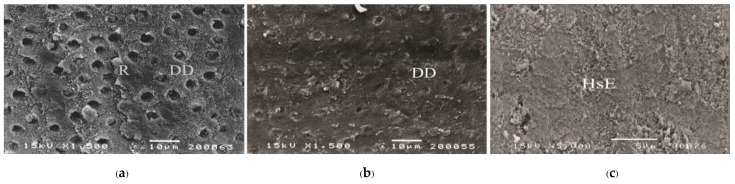
SEM micrograph of the fractured surface showing failure: in the remaining demineralized dentin of Single-Bond2 (**a**) and at the demineralized dentin-resin interface of All-Bond2 (**b**) moist bonding with primer specimens, and in the hybridized suspended smears at the enamel-resin interface of AQ-Bond (**c**) specimen (DD = demineralized dentin, HsE = hybridized suspended enamel smears, R = luting resin).

**Figure 7 polymers-15-00283-f007:**
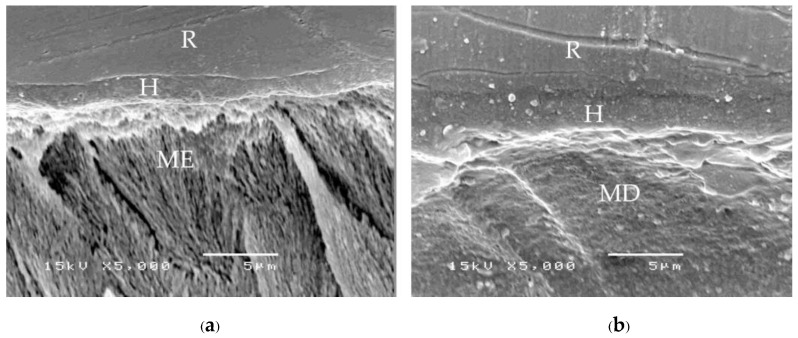
SEM micrographs of fractured specimens after chemical challenge demonstrating: the stable hybridized enamel (**a**) and hybridized dentin (**b**) of 1-1-30s primerless-wet specimens (H = hybrid layer, R = resin, ME = modified enamel, MD = modified dentin).

**Figure 8 polymers-15-00283-f008:**
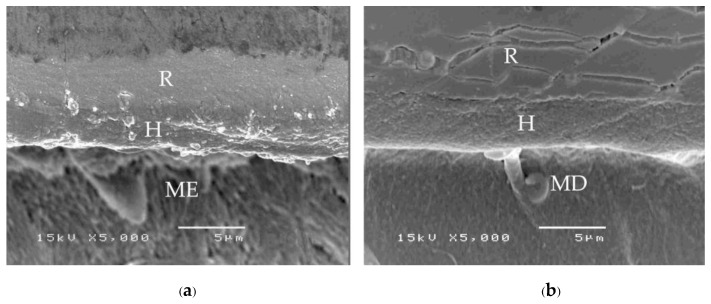
SEM micrographs of fractured specimens after chemical challenge demonstrating: the stable hybridized enamel (**a**) and hybridized dentin (**b**) of Super-Bond C&B specimens (H = hybrid layer, R = resin, ME = modified enamel, MD = modified dentin).

**Figure 9 polymers-15-00283-f009:**
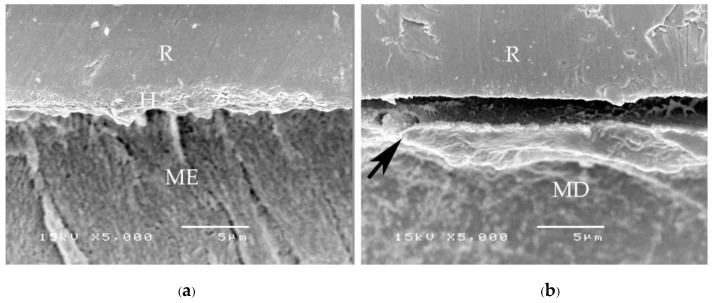
SEM micrographs of fractured specimens after chemical challenge demonstrating: the degraded hybridized enamel (**a**) and the detached and degraded dentin-resin interface (black arrow) of All-Bond2 specimens (**b**) (H = hybrid layer, R = resin, ME = modified enamel, MD = modified dentin).

**Figure 10 polymers-15-00283-f010:**
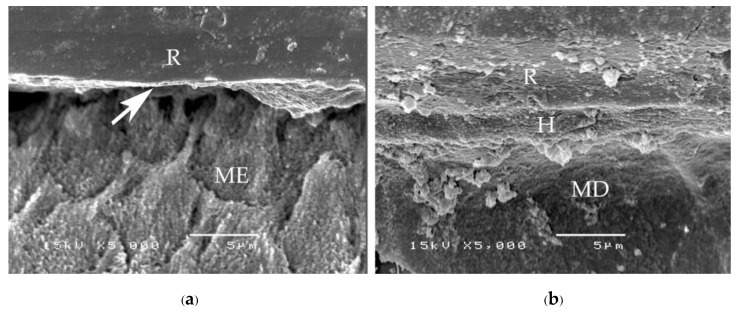
SEM micrograph of fractured specimens after chemical challenge demonstrating: the degraded hybridized enamel (white arrow) (**a**) and the hybridized dentin (**b**) of AQ- Bond specimens (H = hybrid layer, R = resin, ME = modified enamel, MD = modified dentin).

**Table 1 polymers-15-00283-t001:** Manipulation of tooth-conditioning, luting adhesive, and prosthesis cementation.

Systems	Primerless-Wet			Dry	Self-Etched	Moist with Primer	
Groups	1-1-10s	1-1-30s	1-1-60s	Super-Bond C&B	AQ-Bond	All-Bond2	Single-Bond2
Acid conditioner	1-1	1-1	1-1	10-3	-	32% phosphoric acid	35% phosphoric acid
Conditioning time	10 s	30 s	60 s	10 s	-	15 s	15 s
Rinse off	10 s	10 s	10 s	10 s	-	15 s	10 s
Surface treatment	Blot-dried 10 s			Air-dried 10 s	-	Air-dried 1 s, kept moist	Blot-dried and kept moist
Manipulations of luting adhesives	Mixed 4 drops of 4-META/MMA and 1 drop of TBB in a cool porcelain container applied using brush-dip technique with PMMA powder for auto-curing on the conditioned tooth-surface and resin-composite inlay or PMMA block prior fixation			Same as primerless-wet groups	Scrubbed sponge impregnated with monomers on tooth surface for 20 s, air-dried for 5 s, light-cured for 10 s. Applied metafil FLO on resin-composite inlay or PMMA block prior to fixation, light-cured for 60 s.	Mixed 1 drop of primer A: B, coated on conditioned tooth-surface 5 times, gently air-dried for 5 s, applied D&E resin, light-cured for 20 s. Mixed Duolink cement and applied on resin-composite inlay or PMMA block prior to fixation, light-cured for 60 s.	Applied Single-Bond 2 on conditioned tooth surface for 15 s, gently air-dried for 5 s, light cured for 10 s. Mixed RelyX ARC cement and applied on resin-composite inlay or PMMA block before fixation, light-cured 60 s.

**Table 2 polymers-15-00283-t002:** The main chemical composition of luting adhesives and resin composite inlay.

Materials	Chemical Composition
Primerless-wet	Etchant: 1% citric acid and 1% ferric chloride (1-1); water Monomers: 4-methacryloyloxyethyl trimellitate anhydride in methyl methacrylate initiated by tri-*n*-butyl borane (4-META/MMA-TBB) Powder: poly(methyl methacrylate) (PMMA)
Super-Bond C&B	Etchant: 10% citric acid and 3% ferric chloride (10-3); water Monomers: 4-META/MMA-TBB Powder: PMMA
AQ-Bond Plus Metafil FLO	Monomers: methyl methacrylate (MMA); 4-META; urethane dimethacrylate (UDMA); 2-hydroxyethyl methacrylate (HEMA); acetone; water Sponge: polyurethane foam; amine-*p*-toluenesulfonic acid sodium salt (*p*-TSNa) Luting: UDMA; triethylene glycol dimethacrylate (TEGDMA); trimethylolpropane trimethacrylate (TMPT); barium glass
All-Bond2 Duolink	Etchant: 32% phosphoric acid; water Primer: 2% NTG-GMA (N-tolylglycine-glycidyl methacrylate); 16% BPDM (biphenyl dimethacrylate); acetone Bonding: bisphenol A-glycidyl methacrylate (bis-GMA); UDMA, HEMA Luting: bis-GMA; TEGDMA; UDMA; glass filler
Single-Bond2 RelyX ARC	Etchant: 35% phosphoric acid; water Bonding: bis-GMA; HEMA; dimethacrylates, ethanol, water; methacrylate functional copolymer of polyacrylic and polyitaconic acids Luting: bis-GMA; TEGDMA; zirconia/silica filler
Metafil CX	Inlay: UDMA; TEGDMA; TMPT; colloidal silica

**Table 3 polymers-15-00283-t003:** Mean ± SD of dye penetration distance at tooth-resin interface (n = 10), tensile bond strength, and failure modes of enamel/DEJ/dentin-resin dumbbell-shaped specimens (n = 6) for all groups.

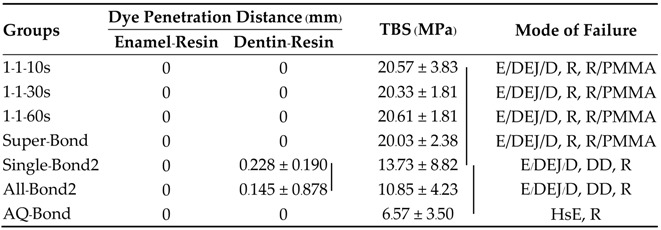

0 = No dye penetration. E/DEJ/D = cohesive failure in enamel, DEJ or dentin, R = cohesive failure in luting resin, R/PMMA = failure at the resin-PMMA-rod interface, DD = failure at demineralized dentin-resin interface, HsE = failure in hybridized suspended enamel smears. There was no significant difference between groups connected with a vertical line (*p* > 0.05).

## Data Availability

Not applicable.

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
