# Peer review of "Marginal Micro-Seal and Tensile Bond Strength of a Biopolymer Hybrid Layer Coupled with Dental Prosthesis Using a Primerless-Wet System"

_polymers, 2023, doi:10.3390/polym15020283_

Round 1
Reviewer 1 Report
- The literature research needs to be improved. More recent references must be used.
- Abbreviations must be explained first and then be used in the manuscript, eg: META, MMA, TBB.
- The author must illustrate the novelty of his work.
- What is the type of used biopolymer?
- The authors must refine the conclusions. In this section, one or more conclusions derived from the study should be included.
Author Response
Dear Reviewer,
The authors would like to thank you for the critical comments and suggestions that helped revise this paper more scientifically. Below is the response to the comments point by point.
- The literature research needs to be improved. More recent references must be used.
Authors: Revised as suggested; added two more recent references (Reference No. 15 and 25)
- Abbreviations must be explained first and then be used in the manuscript, eg: META, MMA, TBB.
Authors: Revised as suggested, except the abstract part due to the limitation of words’ number.
- The author must illustrate the novelty of his work.
Authors: Revised as suggested in the last paragraph of the Discussion section.
- What is the type of used biopolymer?
Authors: The luting adhesives used to fix the dental prosthesis.
- The authors must refine the conclusions. In this section, one or more conclusions derived from the study should be included.
Authors: Revised as suggested
Reviewer 2 Report
The authors are requested to :
1- Revise the English language specially in the introduction chapter.
2- Provide the exact chemical composition of all of the materials used in a table.
3- Provide an illustrated diagram for the steps carried out in their experiment.
4- The authors should modify their hypothesis in the introduction. It is extremely confusing and unclear.
5- The authors should justify how they are comparing different adhesive systems having completely different adhesive monomers and even polymerization mechanisms to each other. It is extremely difficult to point out the factor responsible for the leakage or for improving the bond strength when comparing the reported systems.
Author Response
Dear Reviewer,
The authors would like to thank you for the critical comments and suggestions that helped revise this paper more scientifically. Below is the response to the comments point by point.
1- Revise the English language specially in the introduction chapter.
Authors: Revised as suggested. The manuscript was edited by an American Professor and a British Associate Professor.
2- Provide the exact chemical composition of all of the materials used in a table.
Authors: Revised as suggested (New Table 2).
3- Provide an illustrated diagram for the steps carried out in their experiment.
Authors: Revised as suggested (New Fig.1).
4- The authors should modify their hypothesis in the introduction. It is extremely confusing and unclear.
Authors: Revised as suggested.
5- The authors should justify how they are comparing different adhesive systems having completely different adhesive monomers and even polymerization mechanisms to each other. It is extremely difficult to point out the factor responsible for the leakage or for improving the bond strength when comparing the reported systems.
Authors: Revised as suggested. As the primer and bonding agents of all groups contain the methacrylate monomers with hydrophobic and hydrophilic groups, the significantly different factor is the conditioned tooth surface of each system. This study’s results suggest that the permeability of conditioned tissue of the adhesive system that provides the durable biopolymer hybridized dentin influences the complete micro-seal and higher tensile bond strength.
Reviewer 3 Report
Overall suggestion: Minor changes
1. Has this type of system not been used in the last decade? I notice that all of your references are over 10years old. Is this due to the fact that there has been an evolution of better systems than the ones you are proposing in the last decade? If that is case, how do you justify the need of your system over the existing methods that evolved over the last decade?
2. Could you please include some images or schematics to clearly indicate the steps involved in the resin-composite inlays? Currently, it appears like the materials and methods is lacking this information.
3. Please add a paragraph (7-8 lines) reflecting on the limitations of the study as well as the future directions at the end of discussion to strengthen the article.
Author Response
Dear Reviewer,
The authors would like to thank you for the critical comments and suggestions that helped revise this paper more scientifically. Below is the response to the comments point by point.
- 1. Has this type of system not been used in the last decade? I notice that all of your references are over 10years old. Is this due to the fact that there has been an evolution of better systems than the ones you are proposing in the last decade? If that is case, how do you justify the need of your system over the existing methods that evolved over the last decade?
Authors: No, the results of this proposed system are better than the existing systems. The novelty of this study is that a primerless-wet system using mild acid (1-1) conditioning for 10-60 s and blot-drying to remove all smears and water is less aggressive and safer than a dry bonding system using a 10-3 conditioner. Furthermore, its total etching creates durable hybridized enamel and dentin, providing the micro-seal and tensile bond strength (TBS) better than a moist primer system. In addition, its TBS is higher than the self-etch system. However, an in-vivo study should be carried out to evaluate the effect of dentinal fluid in a vital tooth before introducing this system into the market. In the future, people can use this adhesive system as long-term dentin protection at home.
- 2. Could you please include some images or schematics to clearly indicate the steps involved in the resin-composite inlays? Currently, it appears like the materials and methods is lacking this information.
Authors: Revised as suggested. An illustrated diagram of resin-composite inlays and the fixing steps have been added (New Fig. 1).
- 3. Please add a paragraph (7-8 lines) reflecting on the limitations of the study as well as the future directions at the end of discussion to strengthen the article.
Authors: Revised as suggested. We added a paragraph in the No.1 response at the end of the discussion.
Round 2
Reviewer 2 Report
Please re-revise the English language
Author Response
Dear Reviewer,
The revised manuscript entitled “Marginal micro-seal and tensile bond strength of a biopolymer hybrid layer coupled with dental prosthesis using a primerless-wet system” was re-edited by a Native American Professor.
Best Regards
Corresponding author